# Theoretical Linear Convergence of Unfolded ISTA and its Practical Weights and Thresholds

**Xiaohan Chen**[*]
Department of Computer Science and Engineering
Texas A&M University
College Station, TX 77843, USA
chernxh@tamu.edu

**Jialin Liu**[*]
Department of Mathematics
University of California, Los Angeles
Los Angeles, CA 90095, USA
liujl11@math.ucla.edu

**Zhangyang Wang**
Department of Computer Science and Engineering
Texas A&M University
College Station, TX 77843, USA
atlaswang@tamu.edu

**Wotao Yin**
Department of Mathematics
University of California, Los Angeles
Los Angeles, CA 90095, USA
wotaoyin@math.ucla.edu

## Abstract

In recent years, unfolding iterative algorithms as neural networks has become an empirical success in solving sparse recovery problems. However, its theoretical understanding is still immature, which prevents us from fully utilizing the power of neural networks. In this work, we study unfolded ISTA (Iterative Shrinkage Thresholding Algorithm) for sparse signal recovery. We introduce a weight structure that is necessary for asymptotic convergence to the true sparse signal. With this structure, unfolded ISTA can attain a linear convergence, which is better than the sublinear convergence of ISTA/FISTA in general cases. Furthermore, we propose to incorporate thresholding in the network to perform support selection, which is easy to implement and able to boost the convergence rate both theoretically and empirically. Extensive simulations, including sparse vector recovery and a compressive sensing experiment on real image data, corroborate our theoretical results and demonstrate their practical usefulness. We have made our codes publicly available.[2].

## 1 Introduction

This paper aims to recover a sparse vector $x^*$ from its noisy linear measurements:

$$b = Ax^* + \varepsilon, \qquad (1)$$

where $b \in \mathbb{R}^m$, $x \in \mathbb{R}^n$, $A \in \mathbb{R}^{m \times n}$, $\varepsilon \in \mathbb{R}^m$ is additive Gaussian white noise, and we have $m \ll n$. (1) is an ill-posed, highly under-determined system. However, it becomes easier to solve if $x^*$ is assumed to be sparse, i.e. the cardinality of support of $x^*$, $S = \{i | x_i^* \neq 0\}$, is small compared to $n$.

A popular approach is to model the problem as the LASSO formulation ($\lambda$ is a scalar):

$$\underset{x}{\text{minimize}} \frac{1}{2} \|b - Ax\|_2^2 + \lambda \|x\|_1 \qquad (2)$$

and solve it using iterative algorithms such as the iterative shrinkage thresholding algorithm (ISTA) [1]:

$$x^{k+1} = \eta_{\lambda/L}\Big(x^k + \frac{1}{L}A^T(b - Ax^k)\Big), \quad k = 0, 1, 2, \dots \qquad (3)$$

---

[*]These authors contributed equally and are listed alphabetically.
[2]https://github.com/xchen-tamu/linear-lista-cpss

where $\eta_\theta$ is the soft-thresholding function[3] and $L$ is usually taken as the largest eigenvalue of $A^T A$. In general, ISTA converges sublinearly for any given and fixed dictionary $A$ and sparse code $x^*$ [2]

In [3], inspired by ISTA, the authors proposed a learning-based model named Learned ISTA (LISTA). They view ISTA as a recurrent neural network (RNN) that is illustrated in Figure 1(a), where $W_1 = \frac{1}{L} A^T$, $W_2 = I - \frac{1}{L} A^T A$, $\theta = \frac{1}{L} \lambda$. LISTA, illustrated in Figure 1(b), unrolls the RNN and truncates it into $K$ iterations:

$$x^{k+1} = \eta_{\theta^k}(W_1^k b + W_2^k x^k), \quad k = 0, 1, \cdots, K - 1, \tag{4}$$

leading to a $K$-layer feed-forward neural network with side connections.

Different from ISTA where no parameter is learnable (except the hyper parameter $\lambda$ to be tuned), LISTA is treated as a specially structured neural network and trained using stochastic gradient descent (SGD), over a given training dataset $\{(x_i^*, b_i)\}_{i=1}^N$ sampled from some distribution $\mathcal{P}(x, b)$. All the parameters $\Theta = \{(W_1^k, W_2^k, \theta^k)\}_{k=0}^{K-1}$ are subject to learning. The training is modeled as:

$$\underset{\Theta}{\text{minimize}} \, \mathbb{E}_{x^*, b} \left\| x^K \left( \Theta, b, x^0 \right) - x^* \right\|_2^2. \tag{5}$$

Many empirical results, e.g., [3–7], show that a trained $K$-layer LISTA (with $K$ usually set to $10 \sim 20$) or its variants can generalize more than well to unseen samples $(x', b')$ from the same $\mathcal{P}(x, b)$ and recover $x'$ from $b'$ to the same accuracy within one or two order-of-magnitude fewer iterations than the original ISTA. Moreover, the accuracies of the outputs $\{x^k\}$ of the layers $k = 1, .., K$ gradually improve.

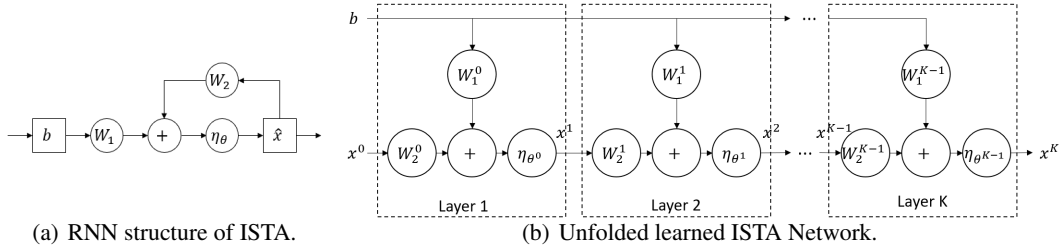

(a) RNN structure of ISTA.  (b) Unfolded learned ISTA Network.

Figure 1: Diagrams of ISTA and LISTA.

## 1.1 Related Works

Many recent works [8, 9, 4, 10, 11] followed the idea of [3] to construct feed-forward networks by unfolding and truncating iterative algorithms, as fast trainable regressors to approximate the solutions of sparse coding models. On the other hand, progress has been slow towards understanding the efficient approximation from a theoretical perspective. The most relevant works are discussed below.

[12] attempted to explain the mechanism of LISTA by re-factorizing the Gram matrix of dictionary, which tries to nearly diagonalize the Gram matrix with a basis that produces a small perturbation of the $\ell_1$ ball. They re-parameterized LISTA into a new factorized architecture that achieved similar acceleration gain to LISTA. Using an "indirect" proof, [12] was able to show that LISTA can converge faster than ISTA, but still sublinearly. Lately, [13] tried to relate LISTA to a projected gradient descent descent (PGD) relying on inaccurate projections, where a trade-off between approximation error and convergence speed was made possible.

[14] investigated the convergence property of a sibling architecture to LISTA, proposed in [4], which was obtained by instead unfolding/truncating the iterative hard thresholding (IHT) algorithm rather than ISTA. The authors argued that they can use data to train a transformation of dictionary that can improve its restricted isometry property (RIP) constant, when the original dictionary is highly correlated, causing IHT to fail easily. They moreover showed it beneficial to allow the weights to decouple across layers. However, the analysis in [14] cannot be straightforwardly extended to ISTA although IHT is linearly convergent [15] under rather strong assumptions.

In [16], a similar learning-based model inspired by another iterative algorithm solve LASSO, approximated message passing (AMP), was studied. The idea was advanced in [17] to substituting the AMP

proximal operator (soft-thresholding) with a learnable Gaussian denoiser. The resulting model, called Learned Denoising AMP (L-DAMP), has theoretical guarantees under the asymptotic assumption named "state evolution." While the assumption is common in analyzing AMP algorithms, the tool is not directly applicable to ISTA. Moreover, [16] shows L-DAMP is MMSE optimal, but there is no result on its convergence rate. Besides, we also note the empirical effort in [18] that introduces an Onsager correction to LISTA to make it resemble AMP.

## 1.2 Motivations and Contributions

We attempt to answer the following questions, which are not fully addressed in the literature yet:

- Rather than training LISTA as a conventional "black-box" network, can we benefit from exploiting certain dependencies among its parameters $\{(W_1^k, W_2^k, \theta^k)\}_{k=0}^{K-1}$ to simplify the network and improve the recovery results?
- Obtained with sufficiently many training samples from the target distribution $\mathcal{P}(x, b)$, LISTA works very well. So, we wonder if there is a theoretical guarantee to ensure that LISTA (4) converges [4] faster and/or produces a better solution than ISTA (3) when its parameters are ideal? If the answer is affirmative, can we quantize the amount of acceleration?
- Can some of the acceleration techniques such as support detection that were developed for LASSO also be used to improve LISTA?

**Our Contributions:** this paper aims to introduce more theoretical insights for LISTA and to further unleash its power. To our best knowledge, this is the first attempt to establish a theoretical convergence rate (upper bound) of LISTA directly. We also observe that the *weight structure* and the *thresholds* can speedup the convergence of LISTA:

- We give a result on asymptotic coupling between the weight matrices $W_1^k$ and $W_2^k$. This result leads us to eliminating one of them, thus reducing the number of trainable parameters. This elimination still retains the theoretical and experimental performance of LISTA.
- ISTA is generally sublinearly convergent before its iterates settle on a support. We prove that, however, there exists a sequence of parameters that makes LISTA converge linearly since its first iteration. Our numerical experiments support this theoretical result.
- Furthermore, we introduce a thresholding scheme for *support selection*, which is extremely simple to implement and significantly boosts the practical convergence. The linear convergence results are extended to support detection with an improved rate.

Detailed discussions of the above three points will follow after Theorems 1, 2 and 3, respectively. Our proofs do not rely on any indirect resemblance, e.g., to AMP [18] or PGD [13]. The theories are supported by extensive simulation experiments, and substantial performance improvements are observed when applying the weight coupling and support selection schemes. We also evaluated LISTA equipped with those proposed techniques in an image compressive sensing task, obtaining superior performance over several of the state-of-the-arts.

## 2 Algorithm Description

We first establish the necessary condition for LISTA convergence, which implies a partial weight coupling structure for training LISTA. We then describe the support-selection technique.

### 2.1 Necessary Condition for LISTA Convergence and Partial Weight Coupling

**Assumption 1** (Basic assumptions). *The signal $x^*$ and the observation noise $\varepsilon$ are sampled from the following set:*

$$(x^*, \varepsilon) \in \mathcal{X}(B, s, \sigma) \triangleq \left\{ (x^*, \varepsilon) \Big| |x_i^*| \leq B, \forall i, \ \|x^*\|_0 \leq s, \|\varepsilon\|_1 \leq \sigma \right\}. \tag{6}$$

*In other words, $x^*$ is bounded and $s$-sparse[5] ($s \geq 2$), and $\varepsilon$ is bounded.*

**Theorem 1** (Necessary Condition). *Given $\{W_1^k, W_2^k, \theta^k\}_{k=0}^{\infty}$ and $x^0 = 0$, let $b$ be observed by (1) and $\{x^k\}_{k=1}^{\infty}$ be generated layer-wise by LISTA (4). If the following holds uniformly for any*

$(x^*, \varepsilon) \in \mathcal{X}(B, s, 0)$ *(no observation noise):*

$$x^k \left( \{W_1^\tau, W_2^\tau, \theta^\tau\}_{\tau=0}^{k-1}, b, x^0 \right) \rightarrow x^*, \quad \text{as } k \rightarrow \infty$$

*and* $\{W_2^k\}_{k=1}^\infty$ *are bounded*

$$\|W_2^k\|_2 \leq B_W, \quad \forall k = 0, 1, 2, \cdots,$$

*then* $\{W_1^k, W_2^k, \theta^k\}_{k=0}^\infty$ *must satisfy*

$$W_2^k - (I - W_1^k A) \rightarrow 0, \quad \text{as } k \rightarrow \infty \tag{7}$$

$$\theta^k \rightarrow 0, \quad \text{as } k \rightarrow \infty. \tag{8}$$

Proofs of the results throughout this paper can be found in the supplementary. The conclusion (7) demonstrates that the weights $\{W_1^k, W_2^k\}_{k=0}^\infty$ in LISTA asymptotically satisfies the following partial weight coupling structure:

$$W_2^k = I - W_1^k A. \tag{9}$$

We adopt the above partial weight coupling for all layers, letting $W^k = (W_1^k)^T \in \Re^{m \times n}$, thus simplifying LISTA (4) to:

$$x^{k+1} = \eta_{\theta^k} \left( x^k + (W^k)^\top (b - Ax^k) \right), \quad k = 0, 1, \cdots, K - 1, \tag{10}$$

where $\{W^k, \theta^k\}_{k=0}^{K-1}$ remain as free parameters to train. Empirical results in Fig. 3 illustrate that the structure (9), though having fewer parameters, improves the performance of LISTA.

The coupled structure (9) for soft-thresholding based algorithms was empirically studied in [16]. The similar structure was also theoretically studied in Proposition 1 of [14] for IHT algorithms using the fixed-point theory, but they let all layers share the same weights, i.e. $W_2^k = W_2, W_1^k = W_1, \forall k$.

## 2.2 LISTA with Support Selection

We introduce a special thresholding scheme to LISTA, called *support selection*, which is inspired by "kicking" [19] in linearized Bregman iteration. This technique shows advantages on recoverability and convergence. Its impact on improving LISTA convergence rate and reducing recovery errors will be analyzed in Section 3. With support selection, at each LISTA layer *before* applying soft thresholding, we will select a certain percentage of entries with largest magnitudes, and trust them as "true support" and won't pass them through thresholding. Those entries that do not go through thresholding will be directly fed into next layer, together with other thresholded entires.

Assume we select $p^k\%$ of entries as the trusted support at layer $k$. LISTA with support selection can be generally formulated as

$$x^{k+1} = \eta_{ss\theta^k}^{p^k} \left( W_1^k b + W_2^k x^k \right), \quad k = 0, 1, \cdots, K - 1, \tag{11}$$

where $\eta_{ss}$ is the thresholding operator with support selection, formally defined as:

$$(\eta_{ss\theta^k}^{p^k}(v))_i = \begin{cases} v_i & : v_i > \theta^k, & i \in S^{p^k}(v), \\ v_i - \theta^k & : v_i > \theta^k, & i \notin S^{p^k}(v), \\ 0 & : -\theta^k \leq v_i \leq \theta^k \\ v_i + \theta^k & : v_i < -\theta^k, & i \notin S^{p^k}(v), \\ v_i & : v_i < -\theta^k, & i \in S^{p^k}(v), \end{cases}$$

where $S^{p^k}(v)$ includes the elements with the largest $p^k\%$ magnitudes in vector $v$:

$$S^{p^k}(v) = \left\{ i_1, i_2, \cdots, i_{p^k} \middle| |v_{i_1}| \geq |v_{i_2}| \geq \cdots |v_{i_{p^k}}| \cdots \geq |v_{i_n}| \right\}. \tag{12}$$

To clarify, in (11), $p^k$ is a hyperparameter to be manually tuned, and $\theta^k$ is a parameter to train. We use an empirical formula to select $p^k$ for layer $k$: $p^k = \min(p \cdot k, p_{\max})$, where $p$ is a positive constant and $p_{\max}$ is an upper bound of the percentage of the support cardinality. Here $p$ and $p_{\max}$ are both hyperparameters to be manually tuned.

If we adopt the partial weight coupling in (9), then (11) is modified as

$$x^{k+1} = \eta_{ss\theta^k}^{p^k} \left( x^k + (W^k)^T (b - Ax^k) \right), \quad k = 0, 1, \cdots, K - 1. \tag{13}$$

**Algorithm abbreviations** For simplicity, hereinafter we will use the abbreviation "CP" for the partial weight coupling in (9), and "SS" for the support selection technique. *LISTA-CP* denotes the LISTA model with weights coupling (10). *LISTA-SS* denotes the LISTA model with support selection (11). Similarly, *LISTA-CPSS* stands for a model using both techniques (13), which has the best performance. Unless otherwise specified, *LISTA* refers to the baseline LISTA (4).

## 3 Convergence Analysis

In this section, we formally establish the impacts of (10) and (13) on LISTA's convergence. The output of the $k^{\text{th}}$ layer $x^k$ depends on the parameters $\{W^\tau, \theta^\tau\}_{\tau=0}^{k-1}$, the observed measurement $b$ and the initial point $x^0$. Strictly speaking, $x^k$ should be written as $x^k\left(\{W^\tau, \theta^\tau\}_{\tau=0}^{k-1}, b, x^0\right)$. By the observation model $b = Ax^* + \varepsilon$, since $A$ is given and $x^0$ can be taken as 0, $x^k$ therefore depends on $\{(W^\tau, \theta^\tau)\}_{\tau=0}^{k}$, $x^*$ and $\varepsilon$. So, we can write $x^k\left(\{W^\tau, \theta^\tau\}_{\tau=0}^{k-1}, x^*, \varepsilon\right)$. For simplicity, we instead just write $x^k(x^*, \varepsilon)$.

**Theorem 2** (Convergence of LISTA-CP). *Given $\{W^k, \theta^k\}_{k=0}^\infty$ and $x^0 = 0$, let $\{x^k\}_{k=1}^\infty$ be generated by (10). If Assumption 1 holds and $s$ is sufficiently small, then there exists a sequence of parameters $\{W^k, \theta^k\}$ such that, for all $(x^*, \varepsilon) \in \mathcal{X}(B, s, \sigma)$, we have the error bound:*

$$\|x^k(x^*, \varepsilon) - x^*\|_2 \le sB \exp(-ck) + C\sigma, \quad \forall k = 1, 2, \cdots, \tag{14}$$

*where $c > 0, C > 0$ are constants that depend only on $A$ and $s$. Recall $s$ (sparsity of the signals) and $\sigma$ (noise-level) are defined in (6).*

If $\sigma = 0$ (noiseless case), (14) reduces to

$$\|x^k(x^*, 0) - x^*\|_2 \le sB \exp(-ck). \tag{15}$$

The recovery error converges to 0 at a linear rate as the number of layers goes to infinity. Combined with Theorem 1, we see that the partial weight coupling structure (10) is both necessary and sufficient to guarantee convergence in the noiseless case. Fig. 3 validates (14) and (15) directly.

**Discussion:** The bound (15) also explains why LISTA (or its variants) can converge faster than ISTA and fast ISTA (FISTA) [2]. With a proper $\lambda$ (see (2)), ISTA converges at an $O(1/k)$ rate and FISTA converges at an $O(1/k^2)$ rate [2]. With a large enough $\lambda$, ISTA achieves a linear rate [20, 21]. With $\bar{x}(\lambda)$ being the solution of LASSO (noiseless case), these results can be summarized as: before the iterates $x^k$ settle on a support[6],

$$x^k \to \bar{x}(\lambda) \text{ sublinearly}, \quad \|\bar{x}(\lambda) - x^*\| = O(\lambda), \quad \lambda > 0$$
$$x^k \to \bar{x}(\lambda) \text{ linearly}, \quad \|\bar{x}(\lambda) - x^*\| = O(\lambda), \quad \lambda \text{ large enough.}$$

Based on the choice of $\lambda$ in LASSO, the above observation reflects an inherent trade-off between convergence rate and approximation accuracy in solving the problem (1), see a similar conclusion in [13]: a larger $\lambda$ leads to faster convergence but a less accurate solution, and vice versa.

However, if $\lambda$ is not constant throughout all iterations/layers, but instead chosen adaptively for each step, more promising trade-off can arise[7]. LISTA and LISTA-CP, with the thresholds $\{\theta^k\}_{k=0}^{K-1}$ free to train, actually adopt this idea because $\{\theta^k\}_{k=0}^{K-1}$ corresponds to a path of LASSO parameters $\{\lambda^k\}_{k=0}^{K-1}$. With extra free trainable parameters, $\{W^k\}_{k=0}^{K-1}$ (LISTA-CP) or $\{W_1^k, W_2^k\}_{k=0}^{K-1}$ (LISTA), learning based algorithms are able to converge to an accurate solution at a fast convergence rate. Theorem 2 demonstrates the existence of such sequence $\{W^k, \theta^k\}_k$ in LISTA-CP (10). The experiment results in Fig. 4 show that such $\{W^k, \theta^k\}_k$ can be obtained by training.

**Assumption 2.** *Signal $x^*$ and observation noise $\varepsilon$ are sampled from the following set:*

$$(x^*, \varepsilon) \in \bar{\mathcal{X}}(B, \underline{B}, s, \sigma) \triangleq \left\{(x^*, \varepsilon) \Big| |x_i^*| \le B, \forall i, \|x^*\|_1 \ge \underline{B}, \|x^*\|_0 \le s, \|\varepsilon\|_1 \le \sigma\right\}. \tag{16}$$

**Theorem 3** (Convergence of LISTA-CPSS). *Given $\{W^k, \theta^k\}_{k=0}^{\infty}$ and $x^0 = 0$, let $\{x^k\}_{k=1}^{\infty}$ be generated by (13). With the same assumption and parameters as in Theorem 2, the approximation error can be bounded for all $(x^*, \varepsilon) \in \mathcal{X}(B, s, \sigma)$:*

$$\|x^k(x^*, \varepsilon) - x^*\|_2 \le sB \exp\Big( -\sum_{t=0}^{k-1} c_{\mathrm{ss}}^t \Big) + C_{\mathrm{ss}}\sigma, \quad \forall k = 1, 2, \cdots, \qquad (17)$$

*where $c_{\mathrm{ss}}^k \ge c$ for all $k$ and $C_{\mathrm{ss}} \le C$.*

*If Assumption 2 holds, $s$ is small enough, and $\underline{B} \ge 2C\sigma$ (SNR is not too small), then there exists another sequence of parameters $\{\tilde{W}^k, \tilde{\theta}^k\}$ that yields the following improved error bound: for all $(x^*, \varepsilon) \in \tilde{\mathcal{X}}(B, \underline{B}, s, \sigma)$,*

$$\|x^k(x^*, \varepsilon) - x^*\|_2 \le sB \exp\Big( -\sum_{t=0}^{k-1} \tilde{c}_{\mathrm{ss}}^t \Big) + \tilde{C}_{\mathrm{ss}}\sigma, \quad \forall k = 1, 2, \cdots, \qquad (18)$$

*where $\tilde{c}_{\mathrm{ss}}^k \ge c$ for all $k$, $\tilde{c}_{\mathrm{ss}}^k > c$ for large enough $k$, and $\tilde{C}_{\mathrm{ss}} < C$.*

The bound in (17) ensures that, with the same assumptions and parameters, LISTA-CPSS is *at least no worse* than LISTA-CP. The bound in (18) shows that, under stronger assumptions, LISTA-CPSS can be *strictly better* than LISTA-CP in both folds: $\tilde{c}_{\mathrm{ss}}^k > c$ is the better convergence rate of LISTA-CPSS; $\tilde{C}_{\mathrm{ss}} < C$ means that the LISTA-CPSS can achieve smaller approximation error than the minimum error that LISTA can achieve.

# 4 Numerical Results

For all the models reported in this section, including the baseline LISTA and LAMP models , we adopt a stage-wise training strategy with learning rate decaying to stabilize the training and to get better performance, which is discussed in the supplementary.

## 4.1 Simulation Experiments

**Experiments Setting.** We choose $m = 250, n = 500$. We sample the entries of $A$ i.i.d. from the standard Gaussian distribution, $A_{ij} \sim N(0, 1/m)$ and then normalize its columns to have the unit $\ell_2$ norm. We fix a matrix $A$ in each setting where different networks are compared. To generate sparse vectors $x^*$, we decide each of its entry to be non-zero following the Bernoulli distribution with $p_b = 0.1$. The values of the non-zero entries are sampled from the standard Gaussian distribution. A test set of 1000 samples generated in the above manner is fixed for all tests in our simulations.

All the networks have $K = 16$ layers. In LISTA models with support selection, we add $p\%$ of entries into support and maximally select $p_{\max}\%$ in each layer. We manually tune the value of $p$ and $p_{\max}$ for the best final performance. With $p_b = 0.1$ and $K = 16$, we choose $p = 1.2$ for all models in simulation experiments and $p_{\max} = 12$ for LISTA-SS but $p_{\max} = 13$ for LISTA-CPSS. The recovery performance is evaluated by NMSE (in dB):

$$\mathrm{NMSE}(\hat{x}, x^*) = 10 \log_{10}\left( \frac{\mathbb{E}\|\hat{x} - x^*\|^2}{\mathbb{E}\|x^*\|^2} \right),$$

where $x^*$ is the ground truth and $\hat{x}$ is the estimate obtained by the recovery algorithms (ISTA, FISTA, LISTA, etc.).

**Validation of Theorem 1.** In Fig 2, we report two values, $\|W_2^k - (I - W_1^k A)\|_2$ and $\theta^k$, obtained by the baseline LISTA model (4) trained under the noiseless setting. The plot clearly demonstrates that $W_2^k \to I - W_1^k A$, and $\theta^k \to 0$, as $k \to \infty$. Theorem 1 is directly validated.

**Validation of Theorem 2.** We report the test-set NMSE of LISTA-CP (10) in Fig. 3. Although (10) fixes the structure between $W_1^k$ and $W_2^k$, the final performance remains the same with the baseline LISTA (4), and outperforms AMP, in both noiseless and noisy cases. Moreover, the output of interior layers in LISTA-CP are even better than the baseline LISTA. In the noiseless case, NMSE converges exponentially to 0; in the noisy case, NMSE converges to a stationary level related with the noise-level. This supports Theorem 2: there indeed exist a sequence of parameters $\{(W^k, \theta^k)\}_{k=0}^{K-1}$ leading to linear convergence for LISTA-CP, and they can be obtained by data-driven learning.

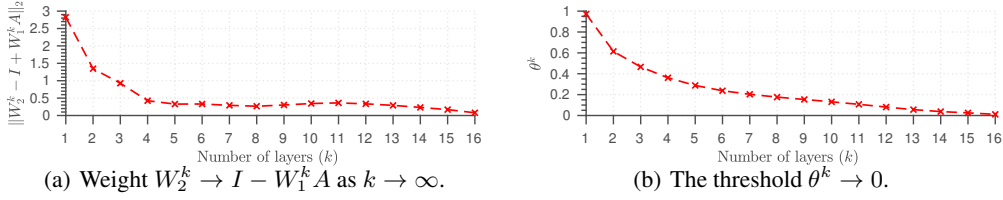

(a) Weight $W_2^k \to I - W_1^k A$ as $k \to \infty$.  (b) The threshold $\theta^k \to 0$.

Figure 2: Validation of Theorem 1.

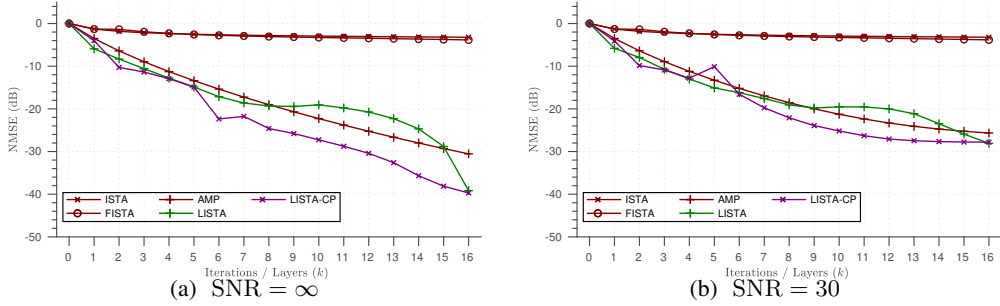

(a) SNR $= \infty$  (b) SNR $= 30$

Figure 3: Validation of Theorem 2.

**Validation of Discussion after Theorem 2.** In Fig 4, We compare LISTA-CP and ISTA with different $\lambda$s (see the LASSO problem (2)) as well as an adaptive threshold rule similar to one in [23], which is described in the supplementary. As we have discussed after Theorem 2, LASSO has an inherent tradeoff based on the choice of $\lambda$. A smaller $\lambda$ leads to a more accurate solution but slower convergence. The adaptive thresholding rule fixes this issue: it uses large $\lambda^k$ for

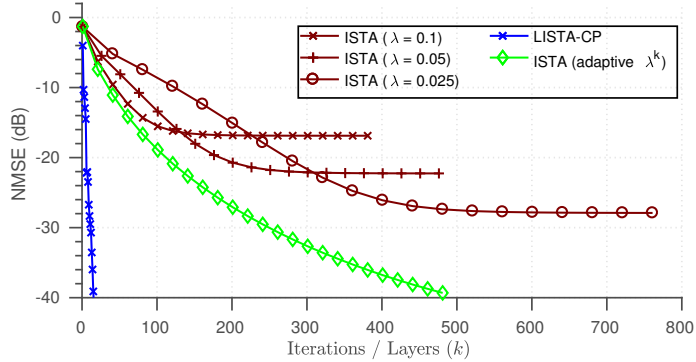

Figure 4: Validating Discussion after Theorem 2 (SNR $= \infty$).

small $k$, and gradually reduces it as $k$ increases to improve the accuracy [23]. Except for adaptive thresholds $\{\theta^k\}_k$ ($\theta^k$ corresponds to $\lambda^k$ in LASSO), LISTA-CP has adaptive weights $\{W^k\}_k$, which further greatly accelerate the convergence. Note that we only ran ISTA and FISTA for 16 iterations, just enough and fair to compare them with the learned models. The number of iterations is so small that the difference between ISTA and FISTA is not quite observable.

**Validation of Theorem 3.** We compare the recovery NMSEs of LISTA-CP (10) and LISTA-CPSS (13) in Fig. 5. The result of the noiseless case (Fig. 5(a)) shows that the recovery error of LISTA-SS converges to 0 at a faster rate than that of LISTA-CP. The difference is significant with the number of layers $k \geq 10$, which supports our theoretical result: "$\tilde{c}_{ss}^k > c$ as $k$ large enough" in Theorem 3. The result of the noisy case (Fig. 5(b)) shows that LISTA-CPSS has better recovery error than LISTA-CP. This point supports $\tilde{C}_{ss} < C$ in Theorem 3. Notably, LISTA-CPSS also outperforms LAMP [16], when $k > 10$ in the noiseless case, and even earlier as SNR becomes lower.

**Performance with Ill-Conditioned Matrix.** We train LISTA, LAMP, LISTA-CPSS with ill-conditioned matrices $A$ of condition numbers $\kappa = 5, 30, 50$. As is shown in Fig. 6, as $\kappa$ increases, the performance of LISTA remains stable while LAMP becomes worse, and eventually inferior to LISTA when $\kappa = 50$. Although our LISTA-CPSS also suffers from ill-conditioning, its performance always stays much better than LISTA and LAMP.

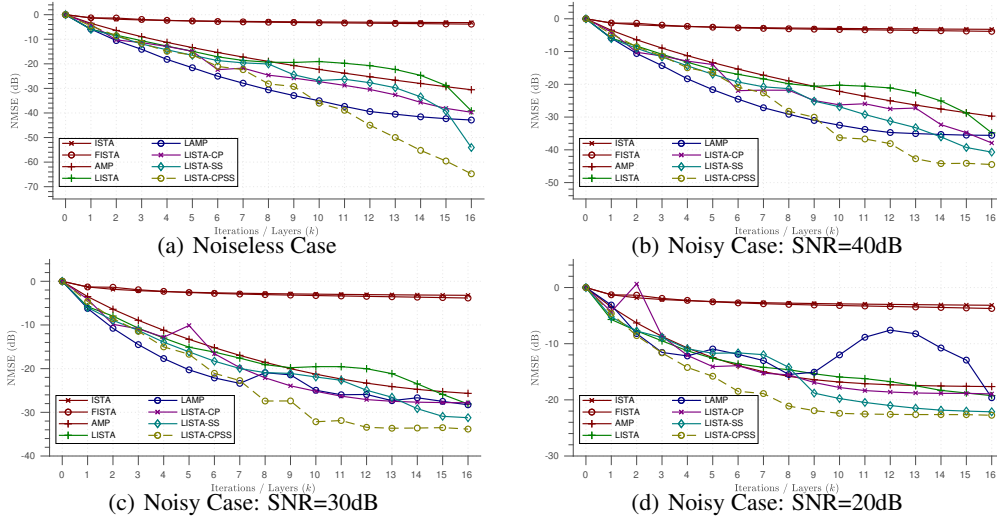

Figure 5: Validation of Theorem 3.

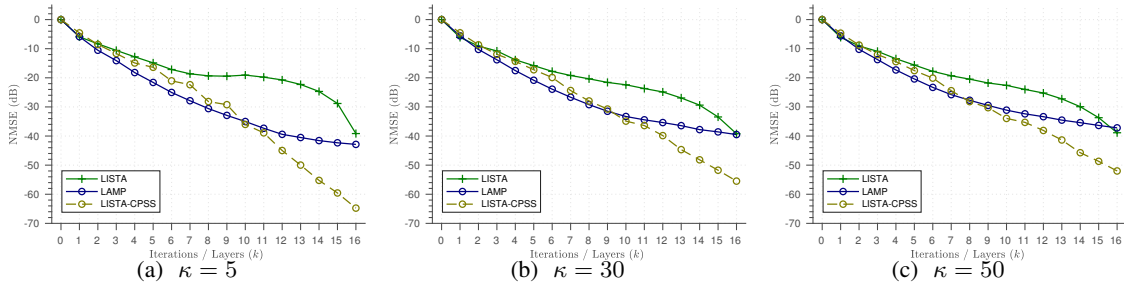

Figure 6: Performance in ill-conditioned situations (SNR = $\infty$).

## 4.2 Natural Image Compressive Sensing

**Experiments Setting.** We perform a compressive sensing (CS) experiment on natural images (patches). We divide the BSD500 [25] set into a training set of 400 images, a validation set of 50 images, and a test set of 50 images. For training, we extract 10,000 patches $f \in \mathbb{R}^{16 \times 16}$ at random positions of each image, with all means removed. We then learn a dictionary $D \in \mathbb{R}^{256 \times 512}$ from them, using a block proximal gradient method [26]. For each testing image, we divide it into non-overlapping $16 \times 16$ patches. A Gaussian sensing matrices $\Phi \in \mathbb{R}^{m \times 256}$ is created in the same manner as in Sec. 4.1, where $\frac{m}{256}$ is the CS ratio.

Since $f$ is typically not exactly sparse under the dictionary $D$, Assumptions 1 and 2 no longer strictly hold. The primary goal of this experiment is thus to show that our proposed techniques remain robust and practically useful in non-ideal conditions, rather than beating all CS state-of-the-arts.

**Network Extension.** In the real data case, we have no ground-truth sparse code available as the regression target for the loss function (5). In order to bypass pre-computing sparse codes $f$ over $D$ on the training set, we are inspired by [11]: first using layer-wise pre-training with a reconstruction loss w.r.t. dictionary $D$ plus an $l_1$ loss, shown in (19), where $k$ is the layer index and $\Theta^k$ denotes all parameters in the $k$-th and previous layers; then appending another learnable fully-connected layer (initialized by $D$) to LISTA-CPSS and perform an end-to-end training with the cost function (20).

$$L^k(\Theta^k) = \sum_{i=1}^{N} \|f_i - D \cdot x_i^k(\Theta^k)\|_2^2 + \lambda \|x_i^k(\Theta^k)\|_1 \tag{19}$$

$$L(\Theta, W_D) = \sum_{i=1}^{N} \|f_i - W_D \cdot x_i^K(\Theta)\|_2^2 + \lambda \|x_i^K(\Theta)\|_1 \tag{20}$$

Table 1: The Average PSRN (dB) for Set 11 test images with CS ratio ranging from 0.2 to 0.6

| Algorithm | 20% | 30% | 40% | 50% | 60% |
|---|---|---|---|---|---|
| TVAL3 | 25.37 | 28.39 | 29.76 | 31.51 | 33.16 |
| Recon-Net | 27.18 | 29.11 | 30.49 | 31.39 | 32.44 |
| LIHT | 25.83 | 27.83 | 29.93 | 31.73 | 34.00 |
| LISTA | 28.17 | 30.43 | 32.75 | 34.26 | 35.99 |
| LISTA-CPSS | **28.25** | **30.54** | **32.87** | **34.60** | **36.39** |

**Results.** The results are reported in Table 1. We build CS models at the sample rates of $20\%, 30\%, 40\%, 50\%, 60\%$ and test on the standard Set 11 images as in [27]. We compare our results with three baselines: the classical iterative CS solver, TVAL3 [28]; the "black-box" deep learning CS solver, Recon-Net [27];a $l_0$-based network unfolded from IHT algorithm [15], noted as LIHT; and the baseline LISTA network, in terms of PSNR (dB)[8]. We build 16-layer LIHT, LISTA and LISTA-CPSS networks and set $\lambda = 0.2$. For LISTA-CPSS, we set $p\% = 0.4\%$ more entries into the support in each layer for support selection. We also select support w.r.t. a percentage of the largest magnitudes within *the whole batch* rather than within a single sample as we do in theorems and simulated experiments, which we emprically find is beneficial to the recovery performance. Table 1 confirms LISTA-CPSS as the best performer among all. The advantage of LISTA-CPSS and LISTA over Recon-Net also endorses the incorporation of the unrolled sparse solver structure into deep networks.

## 5   Conclusions

In this paper, we have introduced a partial weight coupling structure to LISTA, which reduces the number of trainable parameters but does not hurt the performance. With this structure, unfolded ISTA can attain a linear convergence rate. We have further proposed support selection, which improves the convergence rate both theoretically and empirically. Our theories are endorsed by extensive simulations and a real-data experiment. We believe that the methodology in this paper can be extended to analyzing and enhancing other unfolded iterative algorithms.

### Acknowledgments

The work by X. Chen and Z. Wang is supported in part by NSF RI-1755701. The work by J. Liu and W. Yin is supported in part by NSF DMS-1720237 and ONR N0001417121. We would also like to thank all anonymous reviewers for their tremendously useful comments to help improve our work.

## Footnotes

[3]Soft-thresholding function is defined in a component-wise way: $\eta_\theta(x) = \text{sign}(x) \max(0, |x| - \theta)$

[4] The convergence of ISTA/FISTA measures how fast the $k$-th iterate proceeds; the convergence of LISTA measures how fast the output of the $k$-th layer proceeds as $k$ increases.

[5] A signal is $s$-sparse if it has no more than $s$ non-zero entries.

[6]After $x^k$ settles on a support, i.e. as $k$ large enough such that $\mathrm{support}(x^k)$ is fixed, even with small $\lambda$, ISTA reduces to a linear iteration, which has a linear convergence rate [22].

[7]This point was studied in [23, 24] with classical compressive sensing settings, while our learning settings can learn a good path of parameters without a complicated thresholding rule or any manual tuning.

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
