[Supplementary Material]

# Theoretical Linear Convergence of Unfolded ISTA and Its Practical Weights and Thresholds (Supplementary Material)

**Some notation**  For any $n$-dimensional vector $x \in \Re^n$, subscript $x_S$ means the part of $x$ that is supported on the index set $S$:

$$x_S \triangleq [x_{i_1}, x_{i_2}, \cdots, x_{i_{|S|}}]^T, \quad i_1, \cdots, i_{|S|} \in S, \quad i_1 \le i_2 \le \cdots \le i_{|S|},$$

where $|S|$ is the size of set $S$. For any matrix $W \in \Re^{m \times n}$,

$$W(S,S) \triangleq \begin{bmatrix} W(i_1,i_1), W(i_1,i_2), \cdots, W(i_1,i_{|S|}) \\ W(i_2,i_1), W(i_2,i_2), \cdots, W(i_2,i_{|S|}) \\ \cdots \\ W(i_{|S|},i_1), W(i_{|S|},i_2), \cdots, W(i_{|S|},i_{|S|}) \end{bmatrix}, \quad i_1, \cdots, i_{|S|} \in S, \quad i_1 \le i_2 \le \cdots \le i_{|S|},$$

$$W(S,:) \triangleq \begin{bmatrix} W(i_1,1), W(i_1,2), \cdots, W(i_1,n) \\ W(i_2,1), W(i_2,2), \cdots, W(i_2,n) \\ \cdots \\ W(i_{|S|},1), W(i_{|S|},2), \cdots, W(i_{|S|},n) \end{bmatrix}, \quad i_1, \cdots, i_{|S|} \in S, \quad i_1 \le i_2 \le \cdots \le i_{|S|},$$

$$W(:,S) \triangleq \begin{bmatrix} W(1,i_1), W(1,i_2), \cdots, W(1,i_{|S|}) \\ W(2,i_1), W(2,i_2), \cdots, W(2,i_{|S|}) \\ \cdots \\ W(n,i_1), W(n,i_2), \cdots, W(n,i_{|S|}) \end{bmatrix}, \quad i_1, \cdots, i_{|S|} \in S, \quad i_1 \le i_2 \le \cdots \le i_{|S|}.$$

## A  Proof of Theorem 1

*Proof.* By LISTA model (4), the output of the $k$-th layer $x^k$ depends on parameters, observed signal $b$ and initial point $x^0$: $x^k \left( \{W_1^\tau, W_2^\tau, \theta^\tau\}_{\tau=0}^{k-1}, b, x^0 \right)$. Since we assume $(x^*, \varepsilon) \in \mathcal{X}(B, s, 0)$, the noise $\varepsilon = 0$. Moreover, $A$ is fixed and $x^0$ is taken as $0$. Thus, $x^k$ therefore depends on parameters and $x^*$: $x^k \left( \{W_1^\tau, W_2^\tau, \theta^\tau\}_{\tau=0}^{k-1}, x^* \right)$ In this proof, for simplicity, we use $x^k$ denote $x^k \left( \{W_1^\tau, W_2^\tau, \theta^\tau\}_{\tau=0}^{k-1}, x^* \right)$.

**Step 1**  Firstly, we prove $\theta^k \to 0$ as $k \to \infty$.

We define a subset of $\mathcal{X}(B, s, 0)$ given $0 < \tilde{B} \le B$:

$$\tilde{\mathcal{X}}(B, \tilde{B}, s, 0) \triangleq \left\{ (x^*, \varepsilon) \middle| \tilde{B} \le |x_i^*| \le B, \forall i, \|x^*\|_0 \le s, \varepsilon = 0 \right\} \subset \mathcal{X}(B, s, 0).$$

Since $x^k \to x^*$ uniformly for all $(x^*, 0) \in \mathcal{X}(B, s, 0)$, so does for all $(x^*, 0) \in \tilde{\mathcal{X}}(B, B/10, s, 0)$. Then there exists a uniform $K_1 > 0$ for all $(x^*, 0) \in \tilde{\mathcal{X}}(B, B/10, s, 0)$, such that $|x_i^k - x_i^*| < B/10$ for all $i = 1, 2, \cdots, n$ and $k \ge K_1$, which implies

$$\text{sign}(x^k) = \text{sign}(x^*), \quad \forall k \ge K_1. \tag{21}$$

The relationship between $x^k$ and $x^{k+1}$ is

$$x^{k+1} = \eta_{\theta^k} \left( W_2^k x^k + W_1^k b \right).$$

Let $S = \text{support}(x^*)$. Then, (21) implies that, for any $k \ge K_1$ and $(x^*, 0) \in \tilde{\mathcal{X}}(B, B/10, s, 0)$, we have

$$x_S^{k+1} = \eta_{\theta^k} \left( W_2^k(S,S) x_S^k + W_1^k(S,:) b \right).$$

The fact (21) means $x_i^{k+1} \ne 0, \forall i \in S$. By the definition $\eta_\theta(x) = \text{sign}(x) \max(0, |x| - \theta)$, as long as $\eta_\theta(x)_i \ne 0$, we have $\eta_\theta(x)_i = x_i - \theta \, \text{sign}(x_i)$. Thus,

$$x_S^{k+1} = W_2^k(S,S) x_S^k + W_1^k(S,:) b - \theta^k \, \text{sign}(x_S^*).$$

Furthermore, the uniform convergence of $x^k$ tells us, for any $\epsilon > 0$ and $(x^*, 0) \in \tilde{\mathcal{X}}(B, B/10, s, 0)$, there exists a large enough constant $K_2 > 0$ and $\xi_1, \xi_2 \in \Re^{|S|}$ such that $x_S^k = x_S^* + \xi_1, x_S^{k+1} = x_S^* + \xi_2$ and $\|\xi_1\|_2 \leq \epsilon, \|\xi_2\|_2 \leq \epsilon$. Then

$$x_S^* + \xi_2 = W_2^k(S, S)(x_S^* + \xi_1) + W_1^k(S, :)b - \theta^k \operatorname{sign}(x_S^*).$$

Since the noise is supposed to be zero $\varepsilon = 0$, $b = Ax^*$. Substituting $b$ with $Ax^*$ in the above equality, we obtain

$$x_S^* = W_2^k(S, S)x_S^* + W_1^k(S, :)A(:, S)x_S^* - \theta^k \operatorname{sign}(x_S^*) + \xi,$$

where $\|\xi\|_2 = \|W_2^k(S, S)\xi_1 - \xi_2\|_2 \leq (1 + B_W)\epsilon$, $B_W$ is defined in Theorem 1. Equivalently,

$$\left(I - W_2^k(S, S) - W_1^k A(S, S)\right)x_S^* = \theta^k \operatorname{sign}(x_S^*) - \xi. \tag{22}$$

For any $(x^*, 0) \in \tilde{\mathcal{X}}(B/2, B/10, s, 0)$, $(2x^*, 0) \in \tilde{\mathcal{X}}(B, B/10, s, 0)$ holds. Thus, the above argument holds for all $2x^*$ if $(x^*, 0) \in \tilde{\mathcal{X}}(B/2, B/10, s, 0)$. Substituting $x^*$ with $2x^*$ in (22), we get

$$\left(I - W_2^k(S, S) - W_1^k A(S, S)\right)2x_S^* = \theta^k \operatorname{sign}(2x_S^*) - \xi' = \theta^k \operatorname{sign}(x_S^*) - \xi', \tag{23}$$

where $\|\xi'\|_2 \leq (1 + B_W)\epsilon$. Taking the difference between (22) and (23), we have

$$\left(I - W_2^k(S, S) - W_1^k A(S, S)\right)x_S^* = -\xi' + \xi. \tag{24}$$

Equations (22) and (24) imply

$$\theta^k \operatorname{sign}(x_S^*) - \xi = -\xi' + \xi.$$

Then $\theta^k$ can be bounded with

$$\theta^k \leq \frac{3(1 + B_W)}{\sqrt{|S|}}\epsilon, \quad \forall k \geq \max(K_1, K_2). \tag{25}$$

The above conclusion holds for all $|S| \geq 1$. Moreover, as a threshold in $\eta_\theta$, $\theta^k \geq 0$. Thus, $0 \leq \theta^k \leq 3(1 + B_W)\epsilon$ for any $\epsilon > 0$ as long as $k$ large enough. In another word, $\theta^k \to 0$ as $k \to \infty$.

**Step 2**  We prove that $I - W_2^k - W_1^k A \to 0$ as $k \to \infty$.

LISTA model (4) and $b = Ax^*$ gives

$$\begin{aligned}
x_S^{k+1} &= \eta_{\theta^k}\left(W_2^k(S, :)x^k + W_1^k(S, :)b\right) \\
&= \eta_{\theta^k}\left(W_2^k(S, :)x^k + W_1^k(S, :)A(:, S)x_S^*\right) \\
&\in W_2^k(S, :)x^k + W_1^k(S, :)A(:, S)x_S^* - \theta^k \partial\ell_1(x_S^{k+1}),
\end{aligned}$$

where $\partial\ell_1(x)$ is the sub-gradient of $\|x\|_1$. It is a set defined component-wisely:

$$\partial\ell_1(x)_i = \begin{cases} \{\operatorname{sign}(x_i)\} & \text{if } x_i \neq 0, \\ [-1, 1] & \text{if } x_i = 0. \end{cases} \tag{26}$$

The uniform convergence of $x^k$ implies, for any $\epsilon > 0$ and $(x^*, 0) \in \mathcal{X}(B, s, 0)$, there exists a large enough constant $K_3 > 0$ and $\xi_1, \xi_2 \in \Re^n$ such that $x^k = x^* + \xi_3, x^{k+1} = x^* + \xi_4$ and $\|\xi_3\|_2 \leq \epsilon, \|\xi_4\|_2 \leq \epsilon$. Thus,

$$x_S^* + (\xi_4)_S \in W_2^k(S, S)x_S^* + W_2^k(S, :)\xi_3 + W_1^k A(S, S)x_S^* - \theta^k \partial\ell_1(x_S^{k+1})$$

$$\left(I - W_2^k(S, S) - W_1^k A(S, S)\right)x_S^* \in W_2^k(S, :)\xi_3 - (\xi_4)_S - \theta^k \partial\ell_1(x_S^{k+1})$$

By the definition (26) of $\partial\ell_1$, every element in $\partial\ell_1(x), \forall x \in \Re$ has a magnitude less than or equal to 1. Thus, for any $\xi \in \ell_1(x_S^{k+1})$, we have $\|\xi\|_2 \leq \sqrt{|S|}$, which implies

$$\left\|\left(I - W_2^k(S, S) - W_1^k A(S, S)\right)x_S^*\right\|_2 \leq \|W_2^k\|_2\epsilon + \epsilon + \theta^k\sqrt{|S|}.$$

Combined with (25), we obtain the following inequality for all $k \geq \max(K_1, K_2, K_3)$:

$$\left\| \left( I - W_2^k(S, S) - W_1^k A(S, S) \right) x_S^* \right\|_2 \leq \|W_2^k\|_2 \epsilon + \epsilon + 3(1 + B_W)\epsilon = 4(1 + B_W)\epsilon.$$

The above inequality holds for all $(x^*, 0) \in \mathcal{X}(B, s, 0)$, which implies, for all $k \geq \max(K_1, K_2, K_3)$,

$$\sigma_{\max} \left( I - W_2^k(S, S) - W_1^k A(S, S) \right) = \sup_{\substack{\text{support}(x^*) = S \\ \|x_i^*\|_2 = B}} \left\{ \frac{\|(I - W_2^k(S, S) - W_1^k A(S, S))x_S^*\|_2}{B} \right\}$$

$$\leq \sup_{(x^*, 0) \in \mathcal{X}(B, s, 0)} \left\{ \frac{\|(I - W_2^k(S, S) - W_1^k A(S, S))x_S^*\|_2}{B} \right\}$$

$$\leq \frac{4(1 + B_W)}{B} \epsilon.$$

Since $s \geq 2$, $I - W_2^k(S, S) - W_1^k A(S, S) \to 0$ uniformly for all $S$ with $2 \leq |S| \leq s$. Then, $I - W_2^k - W_1^k A \to 0$ as $k \to \infty$. □

# B   Proof of Theorem 2

Before proving Theorem 2, we introduce some definitions and a lemma.

**Definition 1.** *Mutual coherence $\mu$ of $A \in \Re^{m \times n}$ (each column of $A$ is normalized) is defined as:*

$$\mu(A) = \max_{\substack{i \neq j \\ 1 \leq i, j \leq n}} |(A_i)^\top A_j|, \tag{27}$$

*where $A_i$ refers to the $i^{th}$ column of matrix A.*

*Generalized mutual coherence $\tilde{\mu}$ of $A \in \Re^{m \times n}$ (each column of $A$ is normalized) is defined as:*

$$\tilde{\mu}(A) = \inf_{\substack{W \in \Re^{m \times n} \\ (W_i)^T A_i = 1, 1 \leq i \leq n}} \left\{ \max_{\substack{i \neq j \\ 1 \leq i, j \leq n}} |(W_i)^\top A_j| \right\}. \tag{28}$$

The following lemma tells us the generalized mutual coherence is attached at some $\tilde{W} \in \Re^{m \times n}$.

**Lemma 1.** *There exists a matrix $\widetilde{W} \in \Re^{m \times n}$ that attaches the infimum given in (28):*

$$(\widetilde{W_i})^T A_i = 1, 1 \leq i \leq n, \quad \max_{\substack{i \neq j \\ 1 \leq i, j \leq n}} |(\widetilde{W_i})^\top A_j| = \tilde{\mu}$$

*Proof.* Optimization problem given in (28) is a linear programming because it minimizing a piecewise linear function with linear constraints. Since each column of $A$ is normalized, there is at least one matrix in the feasible set:

$$A \in \{W \in \Re^{m \times n} : (W_i)^T A_i = 1, 1 \leq i \leq n\}.$$

In another word, optimization problem (28) is feasible. Moreover, by the definition of infimum bound (28), we have

$$0 \leq \tilde{\mu}(A) \leq \max_{\substack{i \neq j \\ 1 \leq i, j \leq n}} |(A_i)^\top A_j| = \mu(A).$$

Thus, $\tilde{\mu}$ is bounded. According to Corollary 2.3 in [29], a feasible and bounded linear programming problem has an optimal solution. □

Based on Lemma 1, we define a set of "good" weights which $W^k$s are chosen from:

**Definition 2.** *Given $A \in \Re^{m \times n}$, a weight matrix is "good" if it belongs to*

$$\mathcal{X}_W(A) = \operatorname*{arg\,min}_{W \in \Re^{m \times n}} \left\{ \max_{1 \leq i, j \leq n} |W_{i,j}| : (W_i)^T A_i = 1, 1 \leq i \leq n, \max_{\substack{i \neq j \\ 1 \leq i, j \leq n}} |(W_i)^\top A_j| = \tilde{\mu} \right\}. \tag{29}$$

*Let $C_W = \max_{1 \leq i, j \leq n} |W_{i,j}|$, if $W \in \mathcal{X}_W(A)$.*

With definitions (28) and (29), we propose a choice of parameters:

$$W^k \in \mathcal{X}_W(A), \quad \theta^k = \sup_{(x^*, \varepsilon) \in \mathcal{X}(B, s, \sigma)} \{\tilde{\mu}\|x^k(x^*, \varepsilon) - x^*\|_1\} + C_W \sigma, \tag{30}$$

which are uniform for all $(x^*, \varepsilon) \in \mathcal{X}(B, s, \sigma)$. In the following proof line, we prove that (30) leads to the conclusion (14) in Theorem 2.

**Proof of Theorem 2**

*Proof.* In this proof, we use the notation $x^k$ to replace $x^k(x^*, \varepsilon)$ for simplicity.

**Step 1: no false positives.** Firstly, we take $(x^*, \varepsilon) \in \mathcal{X}(B, s, \sigma)$. Let $S = \text{support}(x^*)$. We want to prove by induction that, as long as (30) holds, $x_i^k = 0, \forall i \notin S, \forall k$ (no false positives). When $k = 0$, it is satisfied since $x^0 = 0$. Fixing $k$, and assuming $x_i^k = 0, \forall i \notin S$, we have

$$\begin{aligned}
x_i^{k+1} &= \eta_{\theta^k}\left(x_i^k - \sum_{j \in S}(W_i^k)^T(Ax^k - b)\right) \\
&= \eta_{\theta^k}\left(-\sum_{j \in S}(W_i^k)^T A_j(x_j^k - x_j^*) + (W_i^k)^T \varepsilon\right), \quad \forall i \notin S.
\end{aligned}$$

Since $\theta^k = \tilde{\mu}\sup_{x^*, \varepsilon}\{\|x^k - x^*\|_1\} + C_W \sigma$ and $W^k \in \mathcal{X}_W(A)$,

$$\theta^k \geq \tilde{\mu}\|x^k - x^*\|_1 + C_W\|\varepsilon\|_1 \geq \left|-\sum_{j \in S}(W_i^k)^T A_j(x_j^k - x_j^*) + (W_i^k)^T \varepsilon\right|, \forall i \notin S,$$

which implies $x_i^{k+1} = 0, \forall i \notin S$ by the definition of $\eta_{\theta^k}$. By induction, we have

$$x_i^k = 0, \forall i \notin S, \quad \forall k. \tag{31}$$

In another word, threshold rule in (30) ensures no false positives[9] for all $x^k, k = 1, 2, \cdots$

**Step 2: error bound for one $(x^*, \varepsilon)$.** Next, let's consider the components on $S$. For all $i \in S$,

$$\begin{aligned}
x_i^{k+1} &= \eta_{\theta^k}\left(x_i^k - (W_i^k)^T A_S(x_S^k - x_S^*) + (W_i^k)^T \varepsilon\right) \\
&\in x_i^k - (W_i^k)^T A_S(x_S^k - x_S^*) + (W_i^k)^T \varepsilon - \theta^k \partial\ell_1(x_i^{k+1}),
\end{aligned}$$

where $\partial\ell_1(x)$ is defined in (26). Since $(W_i^k)^T A_i = 1$, we have

$$\begin{aligned}
x_i^k - (W_i^k)^T A_S(x_S^k - x_S^*) &= x_i^k - \sum_{j \in S, j \neq i}(W_i^k)^T A_j(x_j^k - x_j^*) - (x_i^k - x_i^*) \\
&= x_i^* - \sum_{j \in S, j \neq i}(W_i^k)^T A_j(x_j^k - x_j^*).
\end{aligned}$$

Then,

$$x_i^{k+1} - x_i^* \in -\sum_{j \in S, j \neq i}(W_i^k)^T A_j(x_j^k - x_j^*) + (W_i^k)^T \varepsilon - \theta^k \partial\ell_1(x_i^{k+1}), \quad \forall i \in S.$$

By the definition (26) of $\partial\ell_1$, every element in $\partial\ell_1(x), \forall x \in \Re$ has a magnitude less than or equal to 1. Thus, for all $i \in S$,

$$\begin{aligned}
|x_i^{k+1} - x_i^*| &\leq \sum_{j \in S, j \neq i}\left|(W_i^k)^T A_j\right||x_j^k - x_j^*| + \theta^k + |(W_i^k)^T \varepsilon| \\
&\leq \tilde{\mu}\sum_{j \in S, j \neq i}|x_j^k - x_j^*| + \theta^k + C_W\|\varepsilon\|_1
\end{aligned}$$

Equation (31) implies $\|x^k - x^*\|_1 = \|x_S^k - x_S^*\|_1$ for all $k$. Then

$$\|x^{k+1} - x^*\|_1 = \sum_{i \in S} |x_i^{k+1} - x_i^*| \leq \sum_{i \in S} \Big( \tilde{\mu} \sum_{j \in S, j \neq i} |x_j^k - x_j^*| + \theta^k + C_W \sigma \Big)$$

$$= \tilde{\mu}(|S| - 1) \sum_{i \in S} |x_i^k - x_i^*| + \theta^k |S| + |S| C_W \sigma$$

$$\leq \tilde{\mu}(|S| - 1)\|x^k - x^*\|_1 + \theta^k |S| + |S| C_W \sigma$$

**Step 3: error bound for the whole data set.** Finally, we take supremum over $(x^*, \varepsilon) \in \mathcal{X}(B, x, \sigma)$, by $|S| \leq s$,

$$\sup_{x^*, \varepsilon} \{\|x^{k+1} - x^*\|_1\} \leq \tilde{\mu}(s - 1) \sup_{x^*, \varepsilon} \{\|x^k - x^*\|_1\} + s\theta^k + sC_W \sigma.$$

By $\theta^k = \sup_{x^*, \varepsilon} \{\tilde{\mu}\|x^k - x^*\|_1\} + C_W \sigma$, we have

$$\sup_{x^*, \varepsilon} \{\|x^{k+1} - x^*\|_1\} \leq (2\tilde{\mu}s - \tilde{\mu}) \sup_{x^*, \varepsilon} \{\|x^k - x^*\|_1\} + 2sC_W \sigma.$$

By induction, with $c = -\log(2\tilde{\mu}s - \tilde{\mu}), C = \frac{2sC_W}{1 + \tilde{\mu} - 2\tilde{\mu}s}$, we obtain

$$\sup_{x^*, \varepsilon} \{\|x^{k+1} - x^*\|_1\} \leq (2\tilde{\mu}s - \tilde{\mu})^{k+1} \sup_{x^*, \varepsilon} \{\|x^0 - x^*\|_1\} + 2sC_W \sigma \Big( \sum_{\tau=0}^{k+1} (2\tilde{\mu}s - \tilde{\mu})^\tau \Big)$$

$$\leq (2\tilde{\mu}s - \tilde{\mu})^k sB + C\sigma = sB \exp(-ck) + C\sigma.$$

Since $\|x\|_2 \leq \|x\|_1$ for any $x \in \Re^n$, we can get the upper bound for $\ell_2$ norm:

$$\sup_{x^*, \varepsilon} \{\|x^{k+1} - x^*\|_2\} \leq \sup_{x^*, \varepsilon} \{\|x^{k+1} - x^*\|_1\} \leq sB \exp(-ck) + C\sigma.$$

As long as $s < (1 + 1/\tilde{\mu})/2, c = -\log(2\tilde{\mu}s - \tilde{\mu}) > 0$, then the error bound (14) holds uniformly for all $(x^*, \varepsilon) \in \mathcal{X}(B, s, \sigma)$. □

## C   Proof of Theorem 3

*Proof.* In this proof, we use the notation $x^k$ to replace $x^k(x^*, \varepsilon)$ for simplicity.

**Step 1: proving (17).** Firstly, we assume Assumption 1 holds. Take $(x^*, \varepsilon) \in \mathcal{X}(B, s, \sigma)$. Let $S = \text{support}(x^*)$. By the definition of selecting-support operator $\eta_{ss\theta^k}^{p^k}$, using the same argument with the proof of Theorem 2, we have LISTA-CPSS also satisfies $x_i^k = 0, \forall i \notin S, \forall k$ (no false positive) with the same parameters as (30).

For all $i \in S$, by the definition of $\eta_{ss\theta^k}^{p^k}$, there exists $\xi^k \in \Re^n$ such that

$$x_i^{k+1} = \eta_{ss\theta^k}^{p^k} \Big( x_i^k - (W_i^k)^T A_S(x_S^k - x_S^*) + (W_i^k)^T \varepsilon \Big)$$

$$= x_i^k - (W_i^k)^T A_S(x_S^k - x_S^*) + (W_i^k)^T \varepsilon - \theta^k \xi_i^k,$$

where

$$\xi_i^k \begin{cases} = 0 & \text{if } i \notin S \\ \in [-1, 1] & \text{if } i \in S, x_i^{k+1} = 0 \\ = \text{sign}(x_i^{k+1}) & \text{if } i \in S, x_i^{k+1} \neq 0, i \notin S^{p^k}(x^{k+1}), \\ = 0 & \text{if } i \in S, x_i^{k+1} \neq 0, i \in S^{p^k}(x^{k+1}). \end{cases}$$

The set $S^{p^k}$ is defined in (12). Let

$$S^k(x^*, \varepsilon) = \{i | i \in S, x_i^{k+1} \neq 0, i \in S^{p^k}(x^{k+1})\},$$

where $S^k$ depends on $x^*$ and $\varepsilon$ because $x^{k+1}$ depends on $x^*$ and $\varepsilon$. Then, using the same argument with that of LISTA-CP (Theorem 2), we have

$$\|x_S^{k+1} - x_S^*\|_1 \leq \tilde{\mu}(|S| - 1)\|x_S^k - x_S^*\|_1 + \theta^k \big(|S| - |S^k(x^*, \varepsilon)|\big) + |S| C_W \|\varepsilon\|_1.$$

Since $x_i^k = 0, \forall i \notin S$, $\|x^k - x^*\|_2 = \|x_S^k - x_S^*\|_2$ for all $k$. Taking supremum over $(x^*, \varepsilon) \in \mathcal{X}(B, s, \sigma)$, we have

$$\sup_{x^*, \varepsilon} \|x^{k+1} - x^*\|_1 \leq (\tilde{\mu}s - 1) \sup_{x^*, \varepsilon} \|x^k - x^*\|_1 + \theta^k (s - \inf_{x^*, \varepsilon} |S^k(x^*, \varepsilon)|) + sC_W\sigma.$$

By $\theta^k = \sup_{x^*, \varepsilon} \{\tilde{\mu}\|x^k - x^*\|_1\} + C_W\sigma$, we have

$$\sup_{x^*, \varepsilon} \{\|x^{k+1} - x^*\|_1\} \leq \left(2\tilde{\mu}s - \tilde{\mu} - \tilde{\mu} \inf_{x^*, \varepsilon} |S^k(x^*, \varepsilon)|\right) \sup_{x^*, \varepsilon} \{\|x^k - x^*\|_1\} + 2sC_W\sigma.$$

Let

$$c_{\text{ss}}^k = -\log\left(2\tilde{\mu}s - \tilde{\mu} - \tilde{\mu} \inf_{x^*, \varepsilon} |S^k(x^*, \varepsilon)|\right)$$

$$C_{\text{ss}} = 2sC_W \sum_{k=0}^{\infty} \prod_{t=0}^{k} \exp(-c_{\text{ss}}^t)).$$

Then,

$$\sup_{x^*, \varepsilon} \{\|x^k - x^*\|_1\}$$

$$\leq \left(\prod_{t=0}^{k-1} \exp(-c_{\text{ss}}^t)\right) \sup_{x^*, \varepsilon} \{\|x^0 - x^*\|_1\} + 2sC_W \left(\prod_{t=0}^{0} \exp(-c_{\text{ss}}^t)) + \cdots + \prod_{t=0}^{k-1} \exp(-c_{\text{ss}}^t))\right)\sigma$$

$$\leq sB \left(\prod_{t=0}^{k-1} \exp(-c_{\text{ss}}^t)\right) + C_{\text{ss}}\sigma \leq B \exp\left(-\sum_{t=0}^{k-1} c_{\text{ss}}^t\right) + C_{\text{ss}}\sigma.$$

With $\|x\|_2 \leq \|x\|_1$, we have

$$\sup_{x^*, \varepsilon} \{\|x^k - x^*\|_2\} \leq \sup_{x^*, \varepsilon} \{\|x^k - x^*\|_1\} \leq sB \left(\prod_{t=0}^{k-1} \exp(-c_{\text{ss}}^t)\right) + C_{\text{ss}}\sigma.$$

Since $|S^k|$ means the number of elements in $S^k$, $|S^k| \geq 0$. Thus, $c_{\text{ss}}^k \geq c$ for all $k$. Consequently,

$$C_{\text{ss}} \leq 2sC_W \left(\sum_{k=0}^{\infty} \exp(-ck))\right) = 2sC_W \left(\sum_{k=0}^{\infty} (2\tilde{\mu}s - \tilde{\mu})^k\right) = \frac{2sC_W}{1 + \tilde{\mu} - 2\tilde{\mu}s} = C.$$

**Step 2: proving (18).** Secondly, we assume Assumption 2 holds. Take $(x^*, \varepsilon) \in \bar{\mathcal{X}}(B, \underline{B}, s, \sigma)$. The parameters are taken as

$$W^k \in \mathcal{X}_W(A), \quad \theta^k = \sup_{(x^*, \varepsilon) \in \bar{\mathcal{X}}(B, \underline{B}, s, \sigma)} \{\tilde{\mu}\|x^k(x^*, \varepsilon) - x^*\|_1\} + C_W\sigma.$$

With the same argument as before, we get

$$\sup_{(x^*, \varepsilon) \in \bar{\mathcal{X}}(B, \underline{B}, s, \sigma)} \{\|x^k - x^*\|_2\} \leq sB \exp\left(-\sum_{t=0}^{k-1} \tilde{c}_{\text{ss}}^t\right) + \tilde{C}_{\text{ss}}\sigma,$$

where

$$\tilde{c}_{\text{ss}}^k = -\log\left(2\tilde{\mu}s - \tilde{\mu} - \tilde{\mu} \inf_{(x^*, \varepsilon) \in \bar{\mathcal{X}}(B, \underline{B}, s, \sigma)} |S^k(x^*, \varepsilon)|\right) \geq c$$

$$\tilde{C}_{\text{ss}} = 2sC_W \left(\sum_{k=0}^{\infty} \prod_{t=0}^{k} \exp(-\tilde{c}_{\text{ss}}^t))\right) \leq C.$$

Now we consider $S^k$ in a more precise way. The definition of $S^k$ implies

$$|S^k(x^*, \varepsilon)| = \min\left(p^k, \# \text{ of non-zero elements of } x^{k+1}\right). \tag{32}$$

By Assumption 2, it holds that $\|x^*\|_1 \geq \underline{B} \geq 2C\sigma$. Consequently, if $k > 1/c(\log(sB/C\sigma))$, then

$$sB \exp(-ck) + C\sigma < 2C\sigma \leq \|x^*\|_1,$$

which implies

$$\|x^{k+1} - x^*\|_1 \leq sB(\prod_{t=0}^{k} \exp(-\tilde{c}_{ss}^t)) + \tilde{C}_{ss}\sigma \leq sB\exp(-ck) + C\sigma < \|x^*\|_1.$$

Then # of non-zero elements of $x^{k+1} \geq 1$. (Otherwise, $\|x^{k+1} - x^*\|_1 = \|0 - x^*\|_1$, which contradicts.) Moreover, $p^k = \min(pk, s)$ for some constant $p > 0$. Thus, as long as $k \geq 1/p$, we have $p^k \geq 1$. By (32), we obtain

$$|S^k(x^*, \varepsilon)| > 0, \quad \forall k > \max\left(\frac{1}{p}, \frac{1}{c}\log\left(\frac{sB}{C\sigma}\right)\right), \ \forall(x^*, \varepsilon) \in \bar{\mathcal{X}}(B, \underline{B}, s, \sigma).$$

Then, we have $\tilde{c}_{ss}^k > c$ for large enough $k$, consequently, $\tilde{C}_{ss} < C$. □

## D  The adaptive threshold rule used to produce Fig. 4

---
**Algorithm 1:** A thresholding rule for LASSO (Similar to that in [23])

---
**Input**          : Maximum iteration $K$, initial $\lambda^0, \epsilon^0$.
**Initialization**: Let $x^0 = 0, \lambda^1 = \lambda^0, \epsilon^1 = \epsilon^0$.
**1 for** $k = 1, 2, \cdots, K$ **do**
**2**    Conduct ISTA: $x^k = \eta_{\lambda^k/L}\left(x^{k-1} - \frac{1}{L}A^T(Ax^{k-1} - b)\right)$.
**3**    **if** $\|x^k - x^{k-1}\| < \epsilon^k$ **then**
**4**      Let $\lambda^{k+1} \leftarrow 0.5\lambda^k, \epsilon^{k+1} \leftarrow 0.5\epsilon^k$.
**5**    **else**
**6**      Let $\lambda^{k+1} \leftarrow \lambda^k, \epsilon^{k+1} \leftarrow \epsilon^k$.
**7**    **end**
**8 end**
**Output:** $x^K$

---

We take $\lambda^0 = 0.2, \epsilon^0 = 0.05$ in our experiments.

## E  Training Strategy

In this section we have a detailed discussion on the stage-wise training strategy in empirical experiments. Denote $\Theta = \{(W_1^k, W_2^k, \theta^k)\}_{k=0}^{K-1}$ as all the weights in the network. Note that $(W_1^k, W_2^k)$ can be coupled as in (7). Denote $\Theta^\tau = \{(W_1^k, W_2^k, \theta^k)\}_{k=0}^{\tau}$ all the weights in the $\tau$-th and all the previous layers. We assign a learning multiplier $c(\cdot)$, which is initialized as 1, to each weight in the network. Define an initial learning rate $\alpha_0$ and two decayed learning rates $\alpha_1, \alpha_2$. In real training, we have $\alpha_1 = 0.2\alpha_0, \alpha_2 = 0.02\alpha_0$. Our training strategy is described as below:

- Train the network layer by layer. Training in each layer consists of 3 stages.
- In layer $\tau$, $\Theta^{\tau-1}$ is pre-trained. Initialize $c(W_1^\tau), c(W_2^\tau), c(\theta^\tau) = 1$. The actual learning rates of all weights in the following are multiplied by their learning multipliers.
  - Train $(W_1^\tau, W_2^\tau, \theta^\tau)$ the initial learning rate $\alpha_0$.
  - Train $\Theta^\tau = \Theta^{\tau-1} \cup (W_1^\tau, W_2^\tau, \theta^\tau)$ with the learning rates $\alpha_1$ and $\alpha_2$.
- Multiply a decaying rate $\gamma$ (set to 0.3 in experiments) to each weight in $\Theta^\tau$.
- Proceed training to the next layer.

The layer-wise training is widely adopted in previous LISTA-type networks. We add the learning rate decaying that is able to stabilize the training process. It will make the previous layers change very slowly when the training proceeds to deeper layers because learning rates of first several layers will exponentially decay and quickly go to near zero when the training process progresses to deeper layers, which can prevent them varying too far from pre-trained positions. It works well especially

when the unfolding goes deep to $K > 10$. All models trained and reported in experiments section are trained using the above strategy.

**Remark** While adopting the above stage-wise training strategy, we first finish a complete training pass, calculate the intermediate results and final outputs, and then draw curves and evaluate the performance based on these results, instead of logging how the best performance changes when the training process goes deeper. This manner possibly accounts for the reason why some curves plotted in Section 4.1 display some unexpected fluctuations.

## Footnotes

[9]In practice, if we obtain $\theta^k$ by training, but not (30), the learned $\theta^k$ may not guarantee no false positives for all layers. However, the magnitudes on the false positives are actually small compared to those on true positives. Our proof sketch are qualitatively describing the learning-based ISTA.