[Reviews · NeurIPS 2018]

Reviewer 1



This paper proves that the unfolded LISTA, an empirical deep learning solver for sparse codes, can guarantee faster (linear) asymptotical convergence than standard iterative ISTA algorithm. The authors proposed a partial weight coupling structure and a support detection schemes. Rooted in standard LASSO techniques, they are both shown to speed up LISTA convergence. The theories are endorsed by extensive simulations and a real-data CS experiment. Strength: I really like this paper. Simple, elegant, easy-to-implement techniques are backed up by solid theory. Experiments follow a step-by-step manner and accompanies theories fairly well. - ISTA is generally sub-linearly convergent before its iterates settle on a support. Several prior works [8,15,21] show the acceleration effect of LISTA from different views, but this paper for the first time established the linear convergence of LISTA (upper bound). I view that as an important progress in the research direction of NN sparse solvers. - Both weight coupling and support detection are well motivated by theoretical speedup results. They are also very practical and can be “plug-and-play” with standard LISTA, with considerable improvements observed in experiments Weakness: I have no particular argument for weakness. Two suggestions for authors to consider: - How the authors see whether their theory can be extended to convolutional sparse coding, which might be more suitable choices for image CS? - Although IHT and LISTA solve different problems (thus not directly "comparable" in simulation), is it possible that the l0-based networks [4,21] can also achieve competitive performance in real data CS? The authors are suggested to compare in their updated version.

Reviewer 2



This paper introduces improved algorithms for unfolded sparse recovery solvers, provides better theoretical understanding of these algorithms and shows linear convergence (unlike before) for the introduced algorithms. It also does number of experiments showing various properties of the models and improved performance. Paper is nicely done and relevant. It is good and interesting that simplification of the algorithm (having one instead of two trainable matrices) improves the algorithm. - Line 22: Does ISTA and solution to (2) even converge to x* of (1) ? It shouldn’t as the |x| pushes down on the solution. Do you mean to say that it converges sub linearly to the final solution of (2)? In your graphs for ISTA later in the paper, do you do L2 error to x* or to solution of (2). - It would be good to say what sub linearly (or linearly) means - in terms of what? Of course you go into detail in the text later but it would be good to know at this point. - It would be good to state the protocol at the start. We unfold for N steps and train the network to predict (only) at the last step. It is interesting that the algorithm has good predictions at earlier steps, even though it was not trained for that. - 123: and “trust them as true support” -> “trust them as true support and won’t pass them through thresholding”. - Figure 3: How come there is so little difference between ISTA and FISTA - I though the latter converged quite a bit faster. Is it because of my first point?

Reviewer 3



This paper proposes a variant of LISTA with coupled weights and provides a theoretical analysis for it. It shows that if LISTA converges then its weights values and the thresholds in it go to zero. The two variants proposed by the paper are using a combination of soft and hard thresholding and coupling the weights in the network. In the experiments, these changes are shown to give better results. Though the paper is interesting with some theory that is lacking for LISTA, I believe there are several important details that are missing in it: 1. The code used for the proposed technique is very sophisticated with per-layer initialization. It is not clear what code was used for LISTA and LAMP. It might be that the better performance compared to them is just the different optimization used for the method proposed in the paper. 2. Where do the authors use the network extension'' part, where they learn both the dictionary and the network parameters? It is unclear where it is used. 3. In the supplementary material page 2 line 374: I think it should be |S| \ge 1 and not = 1